# The Impact of Both Individual and Contextual Factors on the Acceptance of Personalized Dietary Advice

**DOI:** 10.3390/nu14091866

**Published:** 2022-04-29

**Authors:** Emily P. Bouwman, Machiel J. Reinders, Joris Galama, Muriel C. D. Verain

**Affiliations:** 1Consumer and Chain, Wageningen Economic Research, Wageningen University & Research, 6708 PB Wageningen, The Netherlands; machiel.reinders@wur.nl (M.J.R.); joris.galama@nhlstenden.com (J.G.); muriel.verain@wur.nl (M.C.D.V.); 2Academy of Communication & Creative Business, NHL Stenden University of Applied Sciences, 8917 DD Leeuwarden, The Netherlands

**Keywords:** acceptance, personalized dietary advice, context, mealtime, healthy eating

## Abstract

(1) Background: The aim of the current study is to investigate which between- and within-person factors influence the acceptance of personalized dietary advice. (2) Methods: A repeated measurements design was used in which 343 participants (M (SD) age = 48 (17.3), 49% female) filled out a baseline survey and started with nine repeated surveys. (3) Results: The results show that the acceptance of personalized dietary advice is influenced by both within-person and between-person factors. The acceptance is higher at lunch compared to breakfast and dinner, higher at home than out of home, higher at moments when individuals have a high intention to eat healthily, find weight control an important food choice motive and have a high healthy-eating self-efficacy. Moreover, the acceptance is higher when individuals do not see the eating context as a barrier and when individuals believe that personalized dietary advice has more benefits than risks. (4) Conclusions: Future behavioral interventions that use personalized dietary advice should consider the context as well as individual differences.

## 1. Introduction

The global rise in obesity [1] and non-communicable dietary-related diseases, such as cardiovascular diseases and diabetes [2], emphasize the importance of a shift towards more healthy diets. This is also visible in the large number of behavioral interventions that have been and are being developed in order to promote individuals’ dietary health [3]. Personalized dietary advice is an approach that can support healthy eating behaviors [4] as it uses information on individual characteristics to develop targeted dietary advice, products or services [5]. Individuals with a healthy eating goal can use personalized dietary advice to translate their goals into action. Personalized dietary advice was found to be effective in promoting health behaviors in a recent systematic review of randomized controlled trials [6] and more likely to be read, remembered and viewed as personally relevant than general advice [7,8].

The acceptance of personalized dietary advice is essential for the advice to be effective in promoting healthy dietary behaviors [9]. Previous studies show that both cognitive and affective factors play a role in the acceptance of personalized dietary advice. Cognitive factors, for instance, include the trade-off between perceived benefits and risks of personalized dietary advice [10,11], and affective factors include feelings of ambivalence [9]. Moreover, eating context also plays a role in the acceptance of personalized dietary advice [9]. The importance of considering the context and temporal factors in health psychology is also highlighted in previous work [12,13]. The latter state that the day-to-day variance within each of us (within-person) could be more important for personalized dietary advice than the inter-individual differences that separate people (between-person). However, more research is needed on within-person drivers of the acceptance of personalized dietary advice in addition to between-person drivers. The current study therefore aims to explore how individuals vary regarding their acceptance of personalized dietary advice between one another and within themselves between mealtimes.

### 1.1. Within-Person Drivers of the Acceptance of Personalized Dietary Advice

Literature shows that for behavior change to take place, the following three conditions must be met: the person should be motivated, should be able and have the opportunity to perform the desired behavior. These three factors form the Motivation-Opportunity-Ability (MOA) framework [14] and are also mentioned as important drivers of acceptance of personalized dietary advice [4]. 

Motivation is an important driver of food intake [15] and plays an important role in the acceptance of personalized dietary advice [16]. Goal setting is highlighted as an important factor regarding the acceptance of personalized dietary advice [4]. Previous work investigated the moderating effect of different types of motivation in relation to the acceptance of personalized dietary advice [17]. In that respect, consumers have different motives or motivations for choosing or eating their food [18]. It is found that the health-related food choice motives ‘weight control’ and ‘health’ had a positive effect on the acceptance of personalized dietary advice [19]. Moreover, weight loss is seen as a positive outcome of personalized dietary advice [4]. Furthermore, it is shown that the importance of food choice motives varies at different mealtimes [20,21]. For instance, it is found that health and weight control are least important for snacking and most important at breakfast and/or lunch [21]. It is hypothesized that the acceptance of personalized dietary advice will be higher at moments when individuals are more motivated, i.e., have a high intention to eat healthily (H1) and at moments when individuals find health (H2) and weight control (H3) to be relatively more important food choice motives.

During the day, the opportunity that an individual has to follow personalized dietary advice and to eat healthy foods changes due to different contextual factors [4,9,22]. The context can, for example, be perceived as a barrier to using personalized dietary advice [9,22]. Previous work argues that giving the right advice, at the right moment, while taking the context into account will stimulate the acceptance of personalized dietary advice [23]. A distinction can be made between the social context (eating alone or with others) and the physical context (eating at home or out of home) and both play a role in the acceptance of personalized dietary advice [4]. We expect that eating alone and at home will increase acceptance of personalized dietary advice, as it is argued that it could be difficult to adhere to a personalized diet in social situations [24]. Thus, it is hypothesized that the acceptance of personalized dietary advice is higher when eating at home (H4) and when eating alone (H5).

Investigating the acceptance of personalized dietary advice at different mealtimes gives an idea of how the acceptance changes during the day at moments where individuals make food choices. It can be reasoned that the acceptance of personalized dietary advice is higher at breakfast and lunch compared to dinner, as individuals have more self-regulation earlier in the day which is needed to follow advice [4,25,26]. Moreover, at breakfast and lunch compared to dinner, individuals often eat alone and are often able to choose their own individual meal rather than cooking a meal for others whose wishes also need to be considered. Therefore, it is hypothesized that the acceptance of personalized dietary advice is higher during breakfast and lunch compared to dinner (H6).

Finally, the acceptance of personalized dietary advice is also found to be influenced by an individual’s belief in their ability to perform a desired task (self-efficacy) [27,28]. Following personalized dietary advice requires a certain commitment from individuals, which is likely to be higher for those with higher self-efficacy [4]. An individual’s level of self-efficacy is also expected to change during the day. Self-efficacy is closely related to self-regulation, which is a concept known to change during the day [25,26,29,30] and more likely to be impaired at the end of the day. Therefore, it is hypothesized that the acceptance of personalized dietary advice is higher at moments when individuals have a high healthy-eating self-efficacy (H7).

### 1.2. Between-Person Drivers of the Acceptance of Personalized Dietary Advice

Previous literature on the consumer acceptance of personalized dietary advice focused on several individual drivers of the acceptance of personalized dietary advice.

The acceptance of personalized dietary advice is determined by a decision-making process that weighs the risks and benefits of the personalized approach (risk-benefit perception) [11,28,31,32]. More specifically, these studies indicate that consumers’ acceptance of personalized dietary advice increases with greater perceived benefits and fewer perceived risks associated with personalized dietary advice [10,28]. For example, one study suggested that consumers feel that disclosing their privacy information is too great a risk compared to the benefits offered by personalized dietary advice services [10]. These researchers capture this ‘benefits and risks’ trade-off in an overall information disclosure valuation, called privacy calculus. It is hypothesized that the acceptance of personalized dietary advice is higher when more benefits than risks regarding personalized dietary advice are perceived (H8).

Consumers’ adoption intention regarding personalized dietary advice may not solely be the result of cognitive considerations of benefits and risks. Several studies point out the role of ambivalent feelings in predicting the adoption intention of new technologies and services [9,33]. Ambivalence refers to having both positive and negative feelings simultaneously and may result from conflicting evaluations of benefits and risks [34,35]. As personalized dietary advice may simultaneously stimulate both the urge towards a better health and different barriers such as the fear of data misuse, ambivalent feelings among consumers may very well be the result. It is hypothesized that the acceptance of personalized dietary advice is lower with higher levels of ambivalent feelings towards personalized dietary advice (H9).

The extent to which individuals perceive barriers towards using personalized dietary advice also plays an important role in the acceptance of personalized dietary advice [9]. Social rejection as well as the eating context can both be seen as a barrier to accepting personalized dietary advice [22]. Thus, it is hypothesized that the acceptance of personalized dietary advice is lower for individuals who perceive social rejection as a barrier (H10) as well as for those who perceive the eating context as a barrier (H11).

Finally, demographic differences could also play a role in individual drivers of intention to use personalized dietary advice. For example, females are generally more interested in nutritional health, therefore also more likely to contribute to food research than males and more likely to be interested in personalized dietary advice [6]. Additionally, younger individuals seem more open to change and thus could be more open to personalized dietary advice than older individuals as they might find it harder to adopt something new [6]. Finally, individuals with a lower education seem to be less open to personalized dietary advice than individuals with a higher education [36,37].

### 1.3. Current Research

In sum, the current research focuses on mapping how individuals vary regarding their acceptance of personalized dietary advice between one another and within themselves between mealtimes. More specifically, we investigate whether differences within a person (mealtime, physical context, social context, intention to eat healthily, food choice motives, health and weight control, healthy-eating self-efficacy) and differences between individuals (risk-benefit perception of personalized dietary advice, ambivalent feelings towards personalized dietary advice, perceived barrier eating context, perceived barrier social rejection and demographics) predict differences in acceptance of personalized dietary advice.

## 2. Materials and Methods

### 2.1. Design and Procedure

An observational within-subjects design was applied to investigate the variation in the acceptance of personalized dietary advice. Participants filled out a baseline survey in the last week of August 2021 and filled out nine repeated surveys over a period of three weeks (6 September–24 September 2021). Between-subject predictors were measured at baseline with a 10 min online survey after participants were deemed eligible to participate. Eligible participants had to own a suitable smartphone, be at least 18 years old, not be on vacation and they should eat breakfast, lunch and dinner at least three times a week. When the required sample size for the baseline measurement was reached, participants started with repeated measurements to monitor the within-subject predictors which were sent via SMS (i.e., Short Message Service) on their smartphone. The within-subject predictors were monitored nine times right before a mealtime (i.e., three times a week for three weeks) and every measurement took approximately five minutes to complete (i.e., event-based assessment) [38]. Participants received a push notification to fill out the repeated measurement and the time frame of the measurement was either five or six hours (breakfast: 5 a.m. to 11 a.m.; lunch: 11 a.m. to 16 p.m.; dinner: 16 p.m. to 21 p.m.). To make sure all participants filled out the repeated measurements before a mealtime, they were screened out for a measurement moment when they indicated that they had already eaten their meal or were not going to eat a meal at that moment. Screened out participants were reminded to fill in the survey before the mealtime next time. To control for order effects, participants were randomly divided into six conditions where the order of the three mealtimes varied based on a Latin Square design (see Table 1). 

### 2.2. Sample

A representative sample of a cross-section of the Dutch adult population based on sex, age, education and region was recruited by a professional market research company. In total, 450 eligible Dutch individuals were invited to participate. The final sample of the baseline measurement consisted of 361 participants (response rate = 80.2%), of which 343 filled out at least one repeated measurement and were considered for the analysis. On average, the repeated measurements were filled in 7.1 times (SD = 2.4) and 111 participants filled in all 9 measurements. Based on experience of earlier studies [25,30], we believe this sample size is sufficient to draw meaningful conclusions.

Participants received incentives for their participation in the form of loyalty points which can be exchanged for gift vouchers. To reduce drop-out rates even more, participants received an extra incentive of EUR 10 if they completed the baseline as well as all the nine repeated measurements.

### 2.3. Measurements

#### 2.3.1. Baseline Measurement

Before starting the baseline measurement, all participants were given a definition of personalized dietary advice [9] and a definition of healthy food consumption based on the guidelines of the Dutch Nutrition Centre. Ambivalence towards personalized dietary advice (M = 2.7, SD = 1.3, α = 0.888) was measured with three items on a 7-point semantic differential scale (absolutely does not give me conflicting feelings—gives me a lot of conflicting feelings; absolutely does not give me an uncomfortable feeling—gives me a very uncomfortable feeling; does not give me mixed feelings—gives me strong mixed feelings) [39]. Barriers to using personalized dietary advice were measured regarding the eating context (seven items; M = 4.3, SD = 1.1, α = 0.810) and social rejection (three items: M = 2.6, SD = 1.3, α = 0.834) on a 7-point Likert scale from 1 (Totally disagree) to 7 (Totally agree) [22]. The scale started with the following question: ‘What would stop you from using personalized dietary advice?’. An example of an item that measured barriers in the eating context is: ‘It is hard to follow personalized dietary advice at work’, and an example of an item that measured barriers in the social rejection is: ‘Rejection of personalized dietary advice by my family’. Risk-benefit perception (M = 5.2, SD = 1.2) was measured with one item (Do you think that using personalized dietary advice offers more benefits than risks, or more risks than benefits?) on a 7-point semantic differential scale (more risks—more benefits) [10]. Finally, participants were asked whether they were using personalized services regarding food or physical activity (yes, food services; yes, physical activity services; yes, both; no), with how many people they live, whether they adhere to certain food rules (no; flexitarian; vegetarian; pescatarian; vegan; faith) and if they have a food-related health problem (yes; no).

#### 2.3.2. Repeated Measurement

The social environment was measured by asking with whom they were going to eat (alone: 47.1%; not alone (household, family, friends, colleagues/students, roommates, other): 52.9%), and the physical environment was measured by asking where they were going to eat (at home: 81.4%; out of home (school, work, on the go, catering facility, someone else, hospital, vacation, other): 18.6%). After these questions, all participants were again given the definition of personalized dietary advice [9] and the definition of healthy food consumption based on the guidelines of the Dutch Nutrition Centre. Acceptance of personalized dietary advice (M = 4.4, SD = 1.7) and the intention to eat healthily (M = 5.3, SD = 1.3) were both measured with three items on a 7-point Likert scale from 1 (Totally disagree) to 7 (Totally agree) [28]. The items for acceptance followed the question ‘If I could receive personalized dietary advice, then…’ and were as follows: ‘I would intend to use it at this meal moment’, ‘I would consider to use it at this meal moment’ and ‘I would definitely use it at this meal moment’. The items for intention to eat healthily were the following: ‘I intend to eat healthy at this meal moment’, ‘I consider to eat healthy at this meal moment’ and ‘I will definitely eat healthy at this meal moment’. The food choice motives health (M = 5.1, SD = 1.0) and weight control (M = 4.7, SD = 1.2) were measured with, respectively six and three items on a 7-point Likert scale from 1 (Totally disagree) to 7 (Totally agree) [40]. The scale started with the following question: ‘It is important to me that my [breakfast/lunch/dinner] at this moment…’. The health motives included ‘contains a lot of vitamins and minerals’, ‘keeps me healthy’, ‘is nutritious’, ‘is high in protein’, ‘is good for my skin/teeth/hair/nails, etc.’ and ‘is high in fiber’. The weight control motives included ‘is low in calories’, ‘helps me control my weight’ and ‘is low in fat’. Healthy-eating self-efficacy (M = 5.1, SD = 1.1) was measured with seven items on a 7-point Likert scale from 1 (Totally disagree) to 7 (Totally agree) [41]. The scale started with the following question: ‘At this moment at [breakfast/lunch/dinner]…’. Examples of items are: ‘I am able to consume fruits and vegetables’, ‘I am able to eat a variety of healthy foods to keep my diet balanced’ and ‘I am able to modify recipes to make them healthier’. Finally, the mealtime (breakfast, lunch, dinner) was included in the dataset.

### 2.4. Data Analysis

A multilevel approach was used to analyze the data. A multilevel model is the most suited statistical approach to examine between-person variation and within-person variation in the same model [42]. Moreover, it allows for individuals with missing data points to be analyzed as well, as standard errors are appropriately adjusted [43]. In a multilevel model, the Level 1 predictors consist of the repeated measurements (within-person) and the Level 2 predictors consist of the baseline measurements (between-person). Furthermore, fixed factors are included to capture the average effects across the participants and random factors are included to capture individual differences within the factors. The data was checked for homoscedasticity by comparing residuals to the fitted items in a scatterplot. The plot indicates randomly distributed data, indicating our data is homoscedastic. QQ-plots were used to check normality of residuals of our final model and of the random effects and no radical deviations from normality were found. A correlation matrix was plotted with the numerical independent variables to check for multicollinearity and there is no severe collinearity in our data (no correlations >0.8).

A random intercept and slopes model was estimated using the Maximum Likelihood method with acceptance of personalized dietary advice as the outcome variable, with fixed effects of the following first level predictors: mealtime 1 (lunch, breakfast), mealtime 2 (lunch, dinner), physical environment (at home, out of home), social environment (alone, with others), intention to eat healthily, food choice motive health, food choice motive weight control and healthy-eating self-efficacy, and the following second level predictors: ambivalence towards personalized dietary advice, eating context and social rejection as a barrier to using personalized dietary advice and risk-benefit perception regarding personalized dietary advice, sex (male, female), age and education (low, medium, high). Condition (1 to 6) was not included in the final model, because adding condition to the model, after being dummy coded, increased the BIC value, indicating that the added value of condition does not measure up to the increased complexity of the model. Mealtime and education were dummy coded. Username was included as a random factor, creating a random intercept for every participant. Additionally, the following predictors were added as a random factor because likelihood ratio tests showed that it significantly improved our model: mealtime 1 (lunch, breakfast; variance^mealmoment1^ = 0.07), mealtime 2 (lunch, dinner; variance^mealmoment2^ = 0.15, *p* < 0.001), intention to eat healthily (variance = 0.09, *p* < 0.001), food choice motive health (variance = 0.14, *p* < 0.01), food choice motive weight control (variance = 0.08, *p* < 0.001) and healthy-eating self-efficacy (variance = 0.09, *p* < 0.001). Finally, model fit was examined with a step-down procedure to determine whether adding level 1 and level 2 predictors to the model would improve the model.

The data was analyzed with SPSS version 25 and R version 3.6.1. The R-package lme4-R was used for the random intercept and slopes model.

## 3. Results

### 3.1. Sample

Participants of the baseline questionnaire had a mean age of 48 years (*SD* = 17.3) ranging from 18 to 85, with a sufficient distribution between different age groups (*n*^<35^ = 100; *n*^35–49^ = 91; *n*^50–64^ = 90; *n*^<64^ = 80). Participants consisted of 49% females, and 23.8% had a low education, 42.1% a medium education and 34.1% a higher education. Moreover, about 47.6% of the participants live in the west of the Netherlands and 52.4% in the rest of the Netherlands, which is representative of how many inhabitants live there. 

### 3.2. Within- and between-Individual Differences

As a baseline model, a random intercept model was estimated with acceptance of personalized dietary advice as the outcome variable and username as the random factor to examine the variances between and within individuals. This model shows that acceptance of personalized dietary advice varies within individuals as well as between individuals. A total of 42% of the variance was attributable to within-person differences, while 58% of the variance in the model can be explained by differences between individuals (Intra Class Coefficient = 0.58). The within-person standard deviation shows that on average the acceptance of personalized dietary advice varied 1.09 points on a 7-point scale within individuals. The between-person standard deviation shows that on average the acceptance of personalized dietary advice varied 1.28 points on a 7-point scale between individuals. See Table 2 for an overview.

### 3.3. Within-Person and Between-Person Predictors of the Acceptance of Personalized Dietary Advice

Results of the random intercepts and slopes model (Table 3 and Table 4) show that the acceptance of personalized dietary advice is predicted by several level 1 predictors. Acceptance is predicted to be higher at lunch compared to breakfast (estimate = −0.14, SE = 0.05, *p* < 0.01) and dinner (estimate = −0.12, SE = 0.05, *p* < 0.05), higher at home than out of home (estimate = −0.27, SE = 0.06, *p* < 0.001), higher at moments when the intention to eat healthily is high (estimate = 0.27, SE = 0.03, *p* < 0.001), when weight control is an important food choice motive (estimate = 0.14, SE = 0.04, *p* < 0.001) and when the healthy-eating self-efficacy is high (estimate = 0.27, SE = 0.04, *p* < 0.001). Furthermore, acceptance of personalized dietary advice is also predicted by several level 2 predictors. Acceptance is predicted to be higher when the eating context is not seen as a barrier (estimate = −0.16, SE = 0.06, *p* < 0.01) and when personalized dietary advice is seen to have more benefits than risks (estimate = 0.23, SE = 0.05, *p* < 0.001). The correlations of the random effects show that, for example, when the standard deviation of the intercept increases by one unit, the slope of mealtime1 decreases by 0.39 standard deviations and the slope of the food choice motive weight control decreases by 0.83 standard deviations.

### 3.4. Model Comparison

First a model was estimated without predictors (Model 1), followed by a model with level 1 predictors (Model 2) and finally a model with both level 1 and level 2 predictors (Model 3). Likelihood ratio tests, Akaike information criterion (AIC) and Bayesian information criterion (BIC) show that adding first level predictors to the model results in a significantly improved model (X^2^ = 1022.5, *p* < 0.001). Adding level two predictors also results in a significantly improved model (X^2^ = 40.65, *p* < 0.001); however, the AIC and the BIC value barely changes, and the BIC even increases a bit. The latter indicates that the benefit of adding level 2 predictors does not measure up to the increased complexity of the model. See Table 5 for an overview.

## 4. Discussion

Personalized dietary advice is a promising route towards healthier diets [6]. However, for personalized dietary advice to be effective, it is essential that the advice is accepted by the receiver [9]. Previous work argued that not only inter-individual difference, but also, and probably even more so, day-to-day variances within each of us are important for personalized dietary advice to be effective [13]. The current study aimed to empirically research this matter by investigating the variance in acceptance of personalized dietary advice both within and between persons, and the factors that contribute to this variance.

The results show that indeed both individual characteristics of the receiver and contextual aspects that fluctuate within a person (e.g., mealtime) are important contributing factors to the acceptance of personalized dietary advice. This is in line with previous work which also show the importance of within-individual differences for a variety of psychological mechanisms [13,25,44]. Although the percentage of variance that can be attributed to within-individual differences in acceptance of personalized dietary advice is slightly smaller (42%) than the percentage of variance that can be attributed to between-individual differences, we can still conclude that within-individual differences are highly relevant in understanding the acceptance of personalized dietary advice.

### 4.1. Within-Person Predictors of the Acceptance of Personalized Dietary Advice

An individual’s motivation, opportunity and ability to adhere to a healthy diet at a certain moment are found to be predictive of the person’s acceptance of personalized dietary advice in that moment, thereby showing the relevance of the MOA-framework [14] in understanding the acceptance of personalized dietary advice. In terms of motivation, both the intention to eat healthily and the importance of weight control as a food choice motive are found to be predictive of the acceptance of personalized dietary advice (confirming H1 and H3). However, the importance of health as a food choice motive does not show to be a significant predictor (rejecting H2). This finding could be interpreted by the reasoning of a study that found an indirect effect of the food choice motive health on acceptance of personalized dietary advice and a direct effect of the weight control motive [19]. The authors explain this finding with the idea that individuals who find health important when making a food choice already believe that they have a healthy diet and therefore do not consider that personalized dietary advice would be beneficial for them [19]. Thus, perhaps the food choice motive health is not a predictor of the acceptance of personalized dietary advice, because those individuals for who health is an important food choice motive already eat healthily and therefore do not believe they need personalized dietary advice.

Concerning the opportunity to use a personalized dietary advice, our results show that the acceptance of personalized dietary advice is higher when eating at home compared to out of home (confirming H4). However, the acceptance is not higher when eating alone compared to eating with others, as the social environment did not significantly predict the acceptance of personalized dietary advice (rejecting H5). This finding is surprising, as we expected that personalized dietary advice would be more difficult to adhere to when eating in the company of others, because their wishes have to be considered as well [24]. Though, other health related studies investigating within-person differences also do not find an effect of the social environment. Previous work shows that the social environment is not predictive of an individual’s self-regulation of healthy eating [25] and that the social environment does not impact an individual’s openness to decrease unhealthy foods or increase light products, and openness to increase healthy foods is even lower when eating alone as opposed to eating with others [45]. This indicates that where a meal is being eaten seems to be more important for the acceptance of personalized dietary advice than with who a meal is being eaten.

Regarding mealtime, our findings show that the acceptance of personalized dietary advice is higher during lunch compared to dinner (partly confirming H6), but also higher during lunch compared to breakfast. Moreover, we do not find that the acceptance of personalized dietary advice is higher at breakfast than at dinner (partly rejecting H6). We did not expect to find a difference between breakfast and lunch, as both meals are similar compared to dinner in that they both often consist of a bread meal in the Netherlands and that individuals often choose their own meal rather than cooking a meal for others whose wishes also need to be considered. A possible explanation for a higher acceptance of personalized dietary advice at lunch compared to breakfast is that for most people breakfast is more habitual than lunch, as it is often the same and eaten in the same context [46], while lunch varies more and is also eaten in different contexts such as at work. Stronger habits are harder to change and therefore individuals might be less open to make changes to their breakfast than to their lunch.

Finally, we found that, as expected, the higher an individual’s perceived healthy-eating self-efficacy is at a certain moment, the more personalized dietary advice is accepted (confirming H7). This indicates that individuals are more inclined to use personalized dietary advice at moments when they feel that they are capable of making healthy food choices. This finding supports previous work on the importance of self-efficacy in the shift towards healthy diets [47] and the use of personalized dietary nutrition [4] and adds to it by showing that self-efficacy remains important when contextual influences are considered.

### 4.2. Between-Person Predictors of the Acceptance of Personalized Dietary Advice

Our results show that the acceptance of personalized dietary advice is higher when individuals believe that personalized dietary advice offers more benefits than risks (confirming H8). This so-called privacy calculus seems to be the most important between-person predictor of the acceptance of personalized dietary advice and underpins the relevance of this factor regarding the acceptance of personalized dietary advice [10]. Ambivalent feelings regarding personalized dietary advice are not found to be a predictor of the acceptance of personalized dietary advice (rejecting H9). This is contrary to what was found in another study on the acceptance of personalized dietary advice [9], who found that more ambivalent feelings is related to a lower acceptance. Perhaps the effect of ambivalence did not hold because of the significant role the within-person predictors played in our study. This could indicate that ambivalence does play a role when considering the overall acceptance of personalized dietary advice, but not when considering the acceptance of personalized dietary advice in different contexts. Finally, the more individuals perceive the eating context as a barrier to following up personalized dietary advice, the lower the acceptance of personalized dietary advice is (confirming H11). However, the perception of social rejection as a barrier to performing personalized dietary advice is not related to acceptance (rejecting H10). This is in line with our finding that the social environment does not predict the acceptance of personalized dietary advice. Thus, with who someone is eating (descriptive social norm) and whether others reject personalized dietary advice (injunctive social norm) does not seem important for the acceptance of personalized dietary advice. Perhaps we did not find social norm effects, because participants of our study did not have to actually use personalized dietary advice, as we measured self-reported acceptance of personalized dietary advice, and therefore, they might underestimate the social impact it could have. Another reason could be that accepting personalized dietary advice is something that needs deliberation. Previous work suggests that when a behavior is performed after deliberation, various behavioral, contextual and individual factors can influence the extent to which social norms are effective [48]. Other individuals’ behaviors and beliefs might less likely influence the acceptance of personalized dietary advice, because following personalized dietary advice is something individuals do for themselves, and they do not need the approval of others for that [48].

### 4.3. Implications

This study shows that acceptance of personalized dietary advice differs from person to person, but also varies within a person during the day, across eating contexts. This implies that the characteristics of the eating context should also be taken into account in addition to the characteristics of the person when personalizing dietary advice. More specific, mealtime, the physical environment, intention to eat healthily, weight control as a food choice motive and health eating self-efficacy are important within-person factors to consider. Overall, the findings imply that personalized dietary advice is best accepted at lunch and while eating at home. Personalized dietary advice could therefore best focus on improving lunch at home. Moreover, the results imply that the acceptance of personalized dietary advice will be highest at moments when the receiver is motivated to eat healthily (intention to eat healthily) and feels able to eat healthily (healthy-eating self-efficacy). Personalized dietary advice will therefore be most effective when implemented at moments when individuals are motivated and feel able. Moreover, a service offering personalized dietary advice can help improve both the motivation and ability, for example by communicating the importance of healthy eating and by providing advice that is easy to implement [49]. Finally, the positive relation between weight control as a food choice motive and acceptance of personalized dietary advice implies that the advice has the best chance to succeed at moments when receivers consider their weight in their selection of food, and therefore, it is potentially beneficial to target personalized nutrition advice at mealtimes where individuals want to lose weight. Previous research shows that food choice motives, including health and weight control, are strongly influenced by the context [21,50]. Regarding snacking, weight control was a more important food choice motive for morning and afternoon snacking, while it was a less important motive for late-night snacking [20]. Moreover, it is found that weight control is a more important food choice motive for breakfast and dinner compared to lunch and late-night snacking [20]. This is in line with other work that found that weight control is least important for snacks and most important for breakfast [21].

Next to these context-specific factors, the following two personal factors are important to consider when offering personalized dietary advice: perceptions of barriers in the eating context and perceptions of risks and benefits related to personalized dietary advice. Barriers in the eating context are for example perceived difficulty to follow the advice when eating out of home. This matches with our finding that acceptance is higher at home compared to out of home. Concerning the perceptions of risks and benefits, the results imply that it is important to make potential receivers of personalized dietary advice aware that the benefits of personalized dietary advice outweigh the risks.

### 4.4. Limitations and Future Research

The current study comes with some limitations, resulting in avenues for future research. First of all, the focus of this study was on acceptance of personalized dietary advice, which was a self-reported measurement. Actual behavior was not measured and therefore it cannot be confirmed that the factors we identified as beneficial to the acceptance of personalized dietary advice indeed contribute to a better compliance with personalized dietary advice. Future research should conduct a real-life intervention to test whether personalized dietary advice that is adapted to the context-specific motivation, opportunity and ability of the receiver leads to a better compliance of the advice. It is for example possible that although consumers indicate to accept personalized dietary advice better at lunch compared to breakfast, advice at breakfast may be more effective.

Second, we measured the acceptance of the concept personalized dietary advice in general, while in a real-life situation people would receive a specific advice which they can act upon. Therefore, future research should investigate the acceptance of specific personalized dietary advice in different contexts, as this comes closer to the actual implementation of personalized dietary advice. 

Third, the current study is an observational study. Even though this brings valuable insights about how the acceptance of personalized dietary advice differs between mealtimes, no conclusions can be made about causal relationships. Future research should include experimental designs within a longitudinal study to determine causal effects in a variety of contexts. For instance, randomly implementing personalized dietary advice at breakfast and at lunch and comparing the effectiveness of the advice between these two mealtimes.

Fourth, in the current study, the repeated measurements took place on weekdays. Future research should also investigate the acceptance of personalized nutrition on weekend days, as the need for dietary improvement is generally higher for the weekend than for weekdays [51]. Moreover, individuals could be more open to personalized dietary advice in the weekend, because they generally have more time on the weekend to make changes to their diets and likely deviate from their regular day-to-day habits. On the other hand, the weekend might also be seen as a time to enjoy their meals, and therefore they are less open to using personalized dietary advice for health reasons than during the week. Future research should investigate the acceptance of personalized dietary advice on weekend days compared to weekdays.

Fifth, our study focused on the acceptance of personalized dietary advice regarding healthy diets. However, personalized dietary advice can also be an effective tool to stimulate sustainable diets. The need for more sustainable diets is urgent [52,53,54,55] and recent work shows that plant-based diets have the potential to be both healthy and sustainable at the same time [56,57,58]. Future research should investigate if personalized dietary advice can be used to stimulate sustainable diets, such as plant-based diets.

Sixth, by using the MOA-framework [14], our study provides an overview of how the motivation, opportunity and ability play a role in the acceptance of personalized dietary advice, however, we did not study different types of motivation such as intrinsic and extrinsic motivation [59]. It is possible that the acceptance of personalized dietary advice differs depending on whether someone is intrinsically motivated to eat healthily due to, for instance, a food-related disease [16], or extrinsically motivated due to for instance receiving a discount on healthy foods. Therefore, future research should investigate how different types of motivation affect the acceptance of personalized dietary advice.

Seventh, when interpreting our results, it should be considered that our study took place during the COVID-19 pandemic. Due to the minor COVID-related restrictions that were applied at the time of data collection (e.g., keep 1.5 m distance, work from home when possible, wear a face mask in public places), restaurants being open and recent work that shows that most people did not change their eating behaviors during the pandemic [60,61], we believe the pandemic had little effect on our study. Nevertheless, it could be possible that the percentages of participants who ate at home and alone were slightly higher than it would have been before the COVID-19 pandemic.

## 5. Conclusions

To conclude, this study shows that both contextual (within-person) and individual (between-person) factors contribute to the acceptance of personalized dietary advice. More specifically, acceptance of personalized dietary advice is found to be higher when the person is motivated (intent to eat healthily and is motivated by weight control), has the opportunity (best at lunch and at home) and feels able (high healthy-eating self-efficacy) to comply with the advice. Insights from this study increase our understanding of the acceptance of personalized dietary advice. Providers and developers of personalized nutrition services can benefit from these insights by both considering individual differences and contextual differences in order to make personalized dietary advice better accepted.

## Figures and Tables

**Table 1 nutrients-14-01866-t001:** The order of mealtimes assigned to the 6 conditions using a Latin Square design.

	Week 1			Week 2			Week 3		
Condition	Monday	Tuesday	Thursday	Monday	Tuesday	Thursday	Monday	Tuesday	Thursday
1	Breakfast	Lunch	Dinner	Dinner	Breakfast	Lunch	Lunch	Dinner	Breakfast
2	Breakfast	Dinner	Lunch	Lunch	Breakfast	Dinner	Dinner	Lunch	Breakfast
3	Lunch	Breakfast	Dinner	Dinner	Lunch	Breakfast	Breakfast	Dinner	Lunch
4	Lunch	Dinner	Breakfast	Breakfast	Lunch	Dinner	Dinner	Breakfast	Lunch
5	Dinner	Breakfast	Lunch	Lunch	Dinner	Breakfast	Breakfast	Lunch	Dinner
6	Dinner	Lunch	Breakfast	Breakfast	Dinner	Lunch	Lunch	Breakfast	Dinner

**Table 2 nutrients-14-01866-t002:** The mean and variance components displayed as standard deviations for acceptance of personalized dietary advice.

	Mean	Standard Deviation		Within-Person Variance (%)	Intra Class Coefficient
		Between-Person	Within-Person		
Acceptance of personalized dietary advice	4.4	1.28	1.09	42	0.58

Number of observations = 2435; Groups (Username) = 343.

**Table 3 nutrients-14-01866-t003:** Estimated model fixed effects.

	Self-Regulation				
	Estimate	SE	t	Confidence Intervals	
Fixed effects				2.5%	97.5%
Intercept	−0.04	0.53	0.74	−0.64	1.43
First level					
Mealtime1 (lunch, breakfast) ^a^	−0.14	0.05 **	−2.79	−0.23	−0.04
Mealtime2 (lunch, dinner) ^a^	−0.11	0.05 *	−2.13	−0.22	−0.0
Social Environment (alone, with others) ^b^	−0.01	0.05	−0.29	−0.11	0.08
Physical Environment (at home, out of home) ^c^	−0.28	0.06 ***	−4.80	−0.40	−0.17
Intention to eat healthily	0.26	0.03 ***	8.07	0.20	0.33
Health motive	0.08	0.05	1.70	−0.01	0.18
Weight control motive	0.15	0.04 ***	4.10	0.08	0.23
Healthy-eating self-efficacy	0.27	0.04 ***	7.12	0.19	0.34
Second level					
Ambivalence	−0.03	0.05	−0.58	−0.13	0.07
Eating context as barrier	−0.16	0.06 **	−2.79	−0.26	−0.05
Social rejection as barrier	0.08	0.05	1.54	−0.02	0.18
Risk-benefit Perception	0.21	0.05 ***	4.08	0.11	0.31
Sex (male, female) ^d^	0.04	0.12	0.34	−0.19	0.27
Age	−0.01	0.004 *	−2.24	−0.02	−0.001
Education1 (low, medium) ^e^	−0.03	0.15	−0.18	−0.32	0.27
Education2 (low, high) ^e^	0.01	0.16	0.06	−0.31	0.32

*p*-values estimated via *t*-tests using the Satterthwaite approximations to degrees of freedom; confidence intervals determined with the Wald method. ^a^ Lunch is coded 0, and breakfast and dinner as 1. ^b^ Alone is coded 1, and with others as 2. ^c^ At home is coded 1, and out of home is 2. ^d^ Male is coded as 1, and female as 2. ^e^ Low is coded as 0, medium and high as 1. * *p* < 0.05. ** *p* < 0.01. *** *p* < 0.001.

**Table 4 nutrients-14-01866-t004:** Estimated model random effects.

Grouping	Effect	Variance	SD	Correlation					
Username	Intercept	2.41	1.55						
	Mealtime1 (lunch, breakfast)	0.07	0.26	−0.39					
	Mealtime2 (lunch, dinner)	0.15	0.39	0.11	−0.25				
	Intention to eat healthily	0.09	0.30	−0.04	0.42	0.52			
	Health motive	0.14	0.37	−0.31	−0.16	−0.08	−0.46		
	Weight control motive	0.08	0.28	−0.83	0.64	0.10	0.37	−0.15	
	Healthy-eating self-efficacy	0.09	0.30	0.01	−0.34	−0.77	−0.58	−0.03	−0.32
Residual		0.63	0.79						

**Table 5 nutrients-14-01866-t005:** Model comparison of model 1, 2 and 3.

Model	df	AIC	BIC	Loglik	Deviance	X^2^	X^2^ df	*p*
model 1 (no predictors)	3	8129	8146	−4061	8123			
model 2 (level 1 predictors)	38	7177	7398	−3551	7101	1022.5	35	<0.001
model 3 (level 1 and level 2 predictors)	46	7153	7419	−3530	7061	40.65	8	<0.001

## Data Availability

The datasets generated during and/or analyzed during the current study are available from the corresponding author on reasonable request.

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
