# Peer review of "The Impact of Both Individual and Contextual Factors on the Acceptance of Personalized Dietary Advice"

_nutrients, 2022, doi:10.3390/nu14091866_

Round 1

Reviewer 1 Report

The paper reads to me well, and the method is statistically appropriate. I have a few minor comments.

  1. It would be helpful to have a Table for detailed characteristics (including anthropometric data and disease data, if available) of the study population.
  2. Was anthropometric information collected? Such as weight, height, BMI, etc. If so,
  3. Also it would be helpful to know the disease conditions of the study population, as it could impact the acceptance of personalized dietary advice.
  4. Because this study was conducted during the COVID-19 pandemic, it would also be important to include the pandemic factors, particularly for the current situation. For example, is there any barrier to accept personalized dietary advice because of the pandemic? Did the pandemic impact an individual’s perception of nutrition?

Reviewer 2 Report

Thank you for the opportunity to review this manuscript. The authors have analysed the association between individual and contextual factors, and the acceptance of personalized dietary advice. There are several concerns for the authors to consider.

  1. Participants with a wide range of age (18-85 years) were included in the analysis. This might have reduced the representativeness of the population for some specific groups of age.
  2. Many predictors were included in the analysis based on a relatively limited number of participants. Is the sample size sufficient to test so many predictors? Is multicollinearity tested in the analysis?
  3. Page 7, Line 300: “while 58% of the variance in the model can be explained by differences between individuals (Intra Class Coefficient = .58).” Why is this number not displayed in Table 2?
  4. Table 3: I cannot see in which rows estimates for predictors (Level 1) are located.
  5. Table 5: There are not subheadings for some columns (for example, line 5 [“Intention to eat healthy”], column 7.
